# Water-Saving Soil Conservation Measures Should Be Used in Northern China: Evidence from Runoff Plot Data

Haiyan Fang [1,2]

1    Key Laboratory of Water Cycle and Related Land Surface Processes, Institute of Geographic Sciences and Natural Resources Research, Chinese Academy of Sciences, Beijing 100101, China; fanghy@igsnrr.ac.cn
2    College of Resources and Environment, University of Chinese Academy of Sciences, Beijing 100049, China

**Abstract:** Most of the current studies on soil conservation measures mainly focus on their soil control effect, neglecting their impact on water quantity. In the present study, the latest seven years (2014–2020) of monitored data from 22 runoff plots in the upstream catchment of the Miyun Reservoir, Beijing were used to evaluate the effects of slope, rainfall, and soil conservation measures on soil and water loss, and some implications were given in this water-scarce region. Excluding the impact of soil conservation measures, soil loss increased with the slope gradient and slope length. Runoff and soil loss were greatly affected by the rainfall amount and maximum 30-min rainfall intensity on the bare and cultivated slopes, or by rainfall amount and rainfall duration on almost all of the plots with soil conservation measures. The results indicated that the bare soil suffered the most severe soil loss, with a mean annual soil loss rate (SLR) of 4325 t km$^{-2}$ year$^{-1}$, followed by the cultivated lands without any measure, with an annual SLR of above 3205 t km$^{-2}$ year$^{-1}$. Contour tillage cannot effectively control soil loss on steep slopes. The vegetation measures and terrace, level bench, and fish scale pits, as well as their combinations, can decrease runoff by above 86% and decrease soil loss by 95%, respectively. Water-saving measures should be implemented in the study region. The measures, such as vegetation coverage, terracing, contour tillage, etc., should be carefully implemented on slopes. Bare and cultivated lands should further be implemented with soil conservation measures in this and similar regions in the world.

**Keywords:** runoff; soil loss; rainfall; slope; reduction efficiency

## 1. Introduction

Water erosion is one of the biggest ecological and environmental problems worldwide [1]. It can induce both on-site fertile topsoil and nutrient losses and off-site sediment siltation in river systems, and cause water resource use problems [2–5]. Severe hazards induced by water erosion have been noticed a long time ago, both in developing and developed countries [6,7].

Soil conservation measures, such as contour tillage, terrace, and vegetation plantation, were widely used in the world to combat water erosion, and many studies have been published until very recently. For example, Zhao et al. [8] summarized the runoff plot data on the Chinese Loess Plateau, and pointed out that soil conservation measures had greatly reduced soil and water losses, but not down to background levels (i.e., 500 t km$^{-2}$ year$^{-1}$). Wolka et al. [9] reviewed the effects of soil conservation measures on runoff, soil loss, and crop yields in Sub-Saharan Africa that showed that the effect of crop slope barrier soil conservation techniques on crop yields varied with the rainfall slope. Maetens et al. [10] evaluated the effects of soil conservation measures in reducing plot runoff and soil loss in Europe and the Mediterranean. Xiong et al. [1] also gave a global review of the effects of soil conservation measures on soil loss control. These studies greatly improve our understanding of the effects of soil conservation measures on soil and water losses. However, most of the studies focus on the reduction effects of soil conservation measures, such as

contour tillage, no tillage, horizontal terracing, level ditching, etc., on soil and water losses, considering less their effect on downstream water resource use.

The Miyun Reservoir is one of the most important drinking water sources in Northern China, and provides around 70% of the drinking water for Beijing, the capital city of China [11–14]. Therefore, the water quantity and quality of the reservoir is vitally important. Since the 1980s, especially after 2000, large-scale soil conservation measures, such as terrace, fish scale pit, level bench, contour tillage, vegetation plantation, etc., have been implemented in this region by the government and local farmers [15,16]. In this region, drinking water is the priority for local daily life and economic development [17–19]. Due to the important role of the Miyun Reservoir, many studies have been performed in this region. For example, Qiu et al. [13,14] evaluated the effect of land use management practices on river discharge, indicating that climate variability, especially precipitation and temperature, had great effects on runoff, sediment yield, and nutrient losses, and significantly affected the efficiency of best management practices. Yan et al. [19] also predicted future climate change impacts on streamflow and nitrogen exports into the reservoir. Tang et al. [20] evaluated land use impacts on nonpoint pollution in this catchment, and the results showed that land use change decreased TN and TP by 39.1% and 23.7%, respectively. Some other studies also reported the variations of streamflow and pollutants entering the downstream Miyun Reservoir [21–24]. Regrettably, most of these studies were done at a catchment scale. Furthermore, although the impacts of soil conservation measures, rainfall characteristics, and slope gradient and length on soil loss have thoroughly been documented in the world [8], the effects of different soil conservation measures on soil and water losses from slopes in the Miyiun Reservoir catchment were less studied, with few exceptions [25,26]. Soil conservation measures can not only control soil loss, but also intercept runoff on slopes. If more water was intercepted on slopes by the implemented measures, the water quantity in the downstream reservoir could be reduced, threatening the safety of the drinking water for Beijing. However, this kind of study was done less, although a sharp decrease in water quantity has been found in the Yellow River basin and in the Miyun Reservoir basin [15]. Therefore, did these soil conservation measures influence water resource use? Are they reasonably implemented in this water priority region? Sufficiently understanding the impact of each kind of soil conservation measure on soil and water loss is urgently required to guarantee people's drinking water safety for downstream catchments.

Therefore, runoff and soil loss data from 22 runoff plots in a catchment upstream of the Miyun Reservoir in Beijing were used to (i) identify the characteristics of runoff and soil loss on different slopes, (ii) explore the effects of slope, rainfall characteristics, and soil conservation measures on soil and water losses, and (iii) give some suggestions to reasonably implement soil conservation measures in this region and similar regions of the world. This study can help guide the implementation of soil conservation measures and reduce the loss of drinking water in the study region.

Section 2 describes the study area, runoff plot characteristics, data collection methods, and treatment. Section 3 gives runoff and soil loss characteristics and their responses to rainfall, slope length and gradient, and soil conservation measures. Some suggestions were also given in this section. Section 4 concludes this study.

## 2. Materials and Methods

### 2.1. Description of the Study Area

The present study was done in the Shixia catchment (117°4′30″ E and 40°34′40″ N), which is located upstream of the Miyun Reservoir in Beijing, Northern China (Figure 1). This catchment covers an area of 33 km², ranging from 130 m to 390 m a.s.l. Gneiss is the main lithology, scattered with granite and limestone. This region has a temperate territorial monsoon climate, with a mean annual rainfall amount of 475 mm, above 70% of which occurs in May–September.

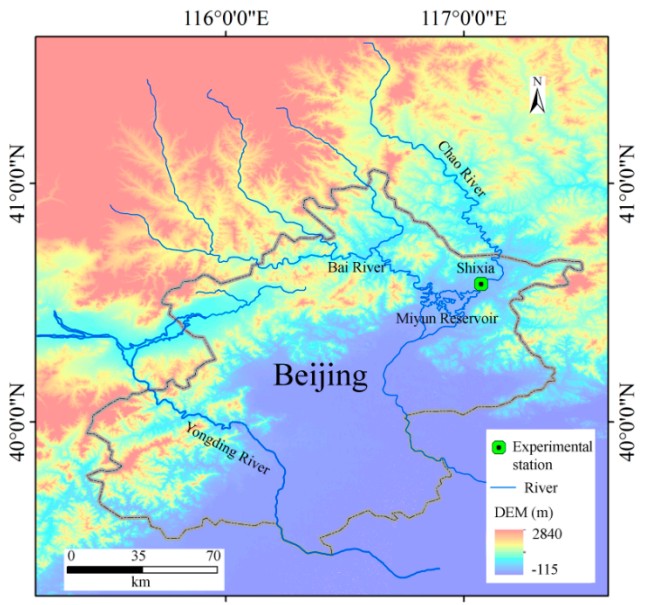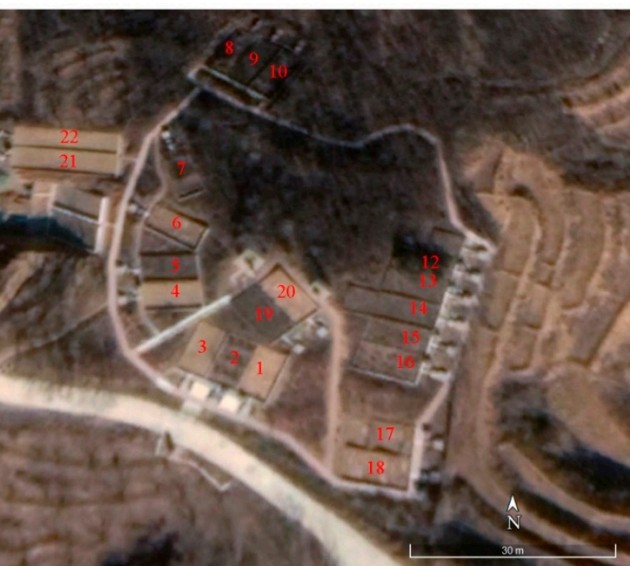

**Figure 1.** Location of the 22 runoff plots. Note: the lower figure was made from a Google image.

In the study region, the main vegetation is *Robinia pseudoacacia, Platycladus orientalis*, and economic forest, and the main crops are corn and wheat. Around 60.9% of the sandy loam soil contains particles larger than 0.05 mm in diameter, 18.22% contains particles 0.05–0.005 mm in diameter, 9.22% contains particles 0.005–0.001 mm in diameter, and 14.88% contains particles less than 0.001 mm in diameter [25].

### 2.2. Runoff Plot and Data Collection

In the Shixia catchment, there were 22 runoff plots in total (Figure 1). The boundaries of each plot were made of bricks and cement. The height of the plot boundary is around 30 cm above ground, with a 40-cm deep insertion into the soils to prevent runoff from leaving or entering the plot. Among the 22 plots, one plot is 5 m long, two plots are 20 m long, and the other plots are 10 m long. The slope degrees of these plots vary from 3.5° to 27°. These plots include five cultivated plots, four bare plots, and thirteen plots with different soil conservation measures. The soil conservation measures implemented on these plots include contour tillage, fish scale pit, level bench, terrace, forest, grass, shrub, and their combinations. Corn was planted on the cultivated plots. Detailed information is provided in Table 1.

In 2014–2020, runoff and sediment discharges were collected with a nine-hole diversion bucket and a tank at the end of each plot. After each rainfall event, the runoff amount was measured. Evenly mixed water and sediment samples were also collected with 1000 mL flasks and transported to a laboratory where sediment concentrations were determined with a dry method [15].

Surface runoff depth (H; mm) for each rainfall event was then calculated with the total rainfall amount and the area of the runoff plot. The event soil loss rate (SLR) was also calculated using the total runoff amount, sediment concentration, and runoff plot area. Annual H and annual SLR were then obtained through summing the events for each plot.

A rain gauge and rain barrel near the plots were used to record rainfall information. In the present study, an erosive rainfall was defined as the one that induced erosion on any of the plots, as proposed by Zhu and Zhu (2014). For each erosive rainfall event, rainfall duration (RD), rainfall amount (P), mean rainfall intensity ($I_m$), and maximum rainfall intensities at 30 min ($I_{30}$) and at 60 min ($I_{60}$) were obtained from rainfall process data.

**Table 1.** Characteristics of the 22 plots. Note: all of the treatments were applied for all seven years (i.e., 2014–2020); the vegetation coverage in the plots was obtained through visual estimation.

| Plot | Slope (°) | Length (m) | Area (m²) | Soil Depth (cm) | Land Use | Soil Conservation Measure |
|------|-----------|------------|-----------|-----------------|----------|---------------------------|
| 1 | 16.5 | 10 | 50 | 30 | Cultivated (corn) | - |
| 2 | 16.5 | 10 | 50 | 30 | Orchard (chestnut) | Terrace (size: 3-m wide) |
| 3 | 16.5 | 10 | 50 | 30 | Bare | - |
| 4 | 14.4 | 10 | 50 | 20 | Bare | - |
| 5 | 14.4 | 10 | 50 | 20 | Shrub | *Vitex negundo* (coverage: 40–60%) |
| 6 | 14.4 | 10 | 50 | 20 | Cultivated (corn) | Contour tillage |
| 7 | 9.4 | 10 | 50 | 20 | Orchard (*hawthorn*) | Terrace (size: 3-m wide) |
| 8 | 27 | 10 | 50 | 20 | Forest and grass | *Robinia* and grass (coverage: 90–100%) |
| 9 | 27 | 10 | 50 | 30 | Shrub | *Vitex negundo* (coverage: 40–50%) |
| 10 | 27 | 10 | 50 | 20 | Forest | *Robinia pseudoacacia* (coverage: 40%); fish-scale pits (size: $1 \times 1.5$ m²) |
| 11 | 19.3 | 5 | 25 | 15 | Shrub | *Vitex negundo* (coverage: 45–60%) |
| 12 | 17.1 | 10 | 50 | 15 | Forest and shrub | *Platycladus orientalis* and *Robinia pseudoacacia* (coverage: 90%); fish-scale pits (size: $1 \times 1.5$ m²) |
| 13 | 18.6 | 10 | 50 | 15 | Shrub | *Vitex negundo* (coverage: 40–50%) |
| 14 | 19.7 | 10 | 50 | 15 | Shrub | *Vitex negundo* (coverage: 40–50%) |
| 15 | 19 | 10 | 50 | 15 | Grass | Alfalfa (coverage: 50–60%) |
| 16 | 19 | 10 | 50 | 15 | Grass | Alfalfa (coverage: 10–15%) |
| 17 | 3.5 | 10 | 50 | 60 | Cultivated (corn) | Terrace (size: 4-m wide) |
| 18 | 3.5 | 10 | 50 | 60 | Cultivated (corn) | Contour tillage |
| 19 | 6.3 | 10 | 50 | 20 | Shrub | *Vitex negundo* (coverage: 45–60%) |
| 20 | 6.3 | 10 | 50 | 20 | Cultivated (corn) | Contour tillage |
| 21 | 14.4 | 20 | 100 | 20 | Bare | - |
| 22 | 14.4 | 20 | 100 | 20 | Bare | - |

*2.3. Data Treatment and Statistical Analysis*

In the present study, Pearson correlation analysis and regression analysis were used to determine the relationships between H, SLR, and the five rainfall eigenvalues. Fisher's protected least significant difference test was also used to compare the means of Hs and SLRs from the plots. Treatments were considered significant if $p < 0.05$. All of the statistics were conducted using SPSS version 14.0 for Windows.

**3. Results and Discussion**

*3.1. Rainfall Characteristics*

The mean annual Ps in 2014–2020 ranged from 385 mm in 2020 to 532.2 mm in 2018, with an average of 475.1 mm, a coefficient of variation of 0.11, and a skewness of −0.86. The annual erosive Ps varied from 239.2 mm in 2020 to 434.7 mm in 2018, with an average of 297.1 mm, or 62.3% of the annual mean P (Figure 2 and Table 2). In 2014–2020, over 50% of the annual Ps produced runoff on the plots. These percentages were higher than those on the Chinese Loess Plateau reported by Liang et al. (2020).

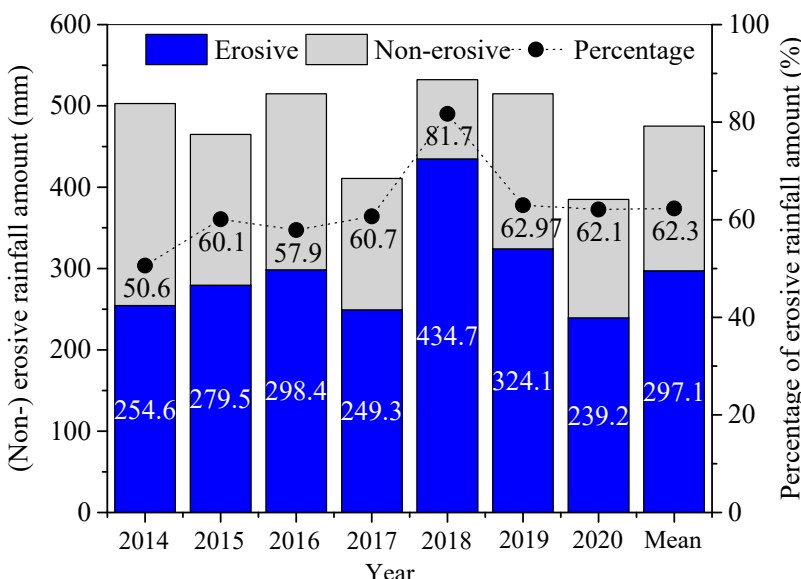

**Figure 2.** Annual erosive and non-erosive rainfall and the percentages of erosive rainfall amounts in 2014–2020.

**Table 2.** Statistical characteristics of annual (erosive) rainfall and the percent of erosive to total rainfall amount in 2014–2020.

|  | Annual P (mm) | Annual Erosive P (mm) | Percent of Mean Annual Erosive Rainfall Amount to the Annual Mean P (%) |
|---|---|---|---|
| Mean | 475.1 | 297.1 | 62.3 |
| Range | 385.0–532.2 | 239.1–434.7 | 50.6–81.7 |
| Std. | 52.9 | 62.6 | 8.8 |
| CV | 0.11 | 0.21 | 0.14 |
| Skewness | -0.86 | 1.74 | 1.57 |

In 2014–2020, there were 66 erosive rainfall events. The event Ps ranged from 4.8 to 108.1mm, with an average of 31.5 mm. The Ps in the range of 10–20 mm occupied over one third of the total number of rainfall events. All of these erosive rainfall events ranged from 0.3 to 32.3 h in RD, with an average of 6.9 h, from 1.0 to 54.0 mm h$^{-1}$ in $I_m$ with an average of 9.5 mm h$^{-1}$, and from 3.6 to 61.7 mm h$^{-1}$ and 6.0 to 64.2 mm h$^{-1}$ in $I_{60}$ and $I_{30}$, respectively (Figure 3). Under future climate change, more extreme rainfall events could occur [13], inducing more severe soil erosion in the study region.

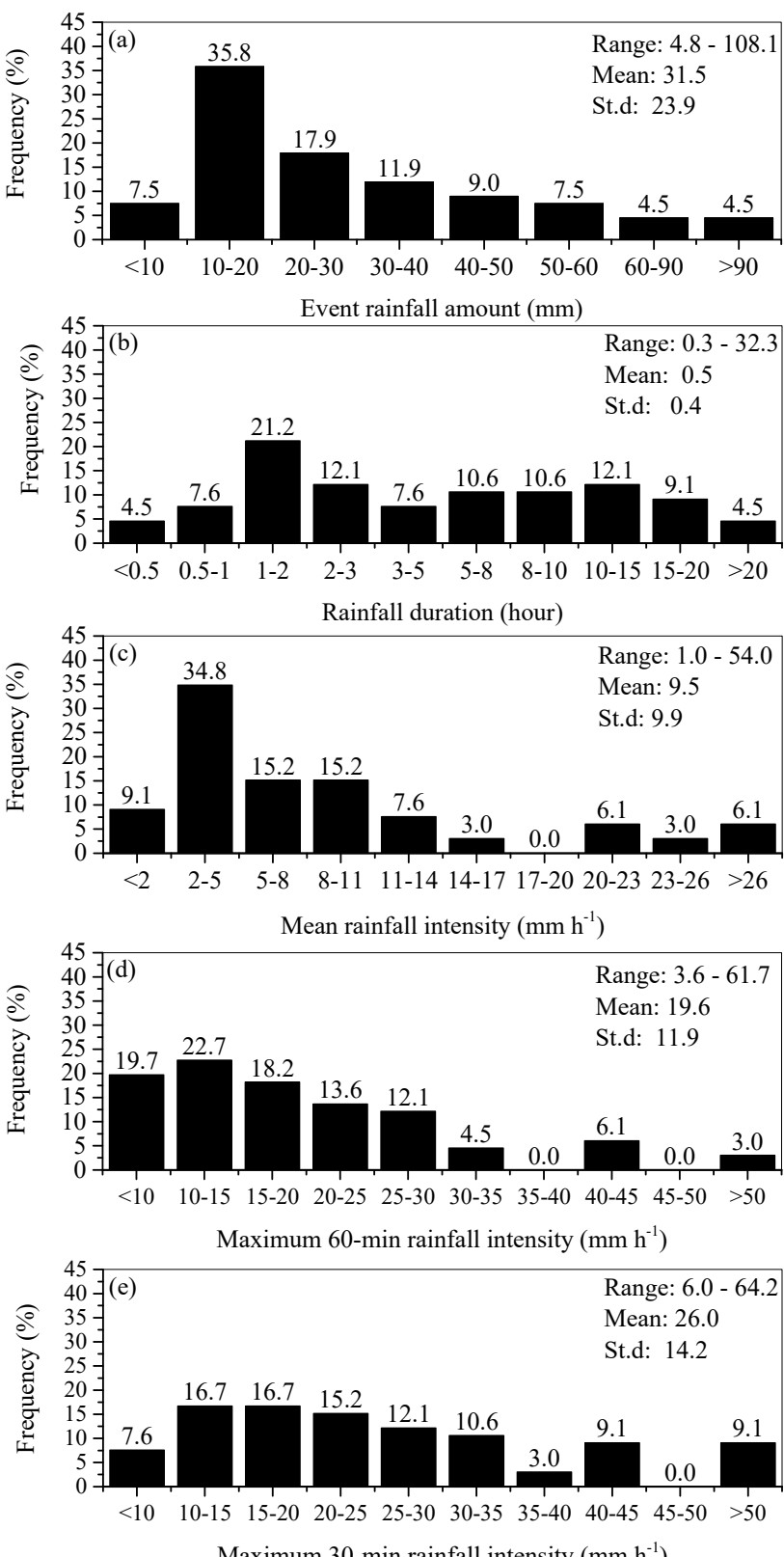

**Figure 3.** Frequencies of some eigenvalues of the erosive rainfall events between 2014–2020: (**a**) event rainfall amount, (**b**) rainfall duration, (**c**) mean rainfall intensity, (**d**) maximum 60-min rainfall intensity, and (**e**) maximum 30-min rainfall intensity.

### 3.2. Runoff and Soil Loss

Annual and event Hs varied widely among the 22 runoff plots (Figure 4a and Table 3). The mean annual Hs in 2014–2020 ranged from 0 on plots 8 and 10 to 100.61 mm on plot 22. The mean annual Hs were less than 18.03 mm on most of the plots, which were significantly less than those on the bare and cultivated plots 1, 3, 4, 6, 21, and 22. Event Hs also differed greatly. The maximum event H was 10.67 mm on plot 22, and the minimum H was zero on plots 8 and 10 (Table 3). Most of the plots had a mean event H less than 2.00 mm. The event Hs on plots 1, 3, 4, 6, 21, and 22 were significantly higher than those on other plots.

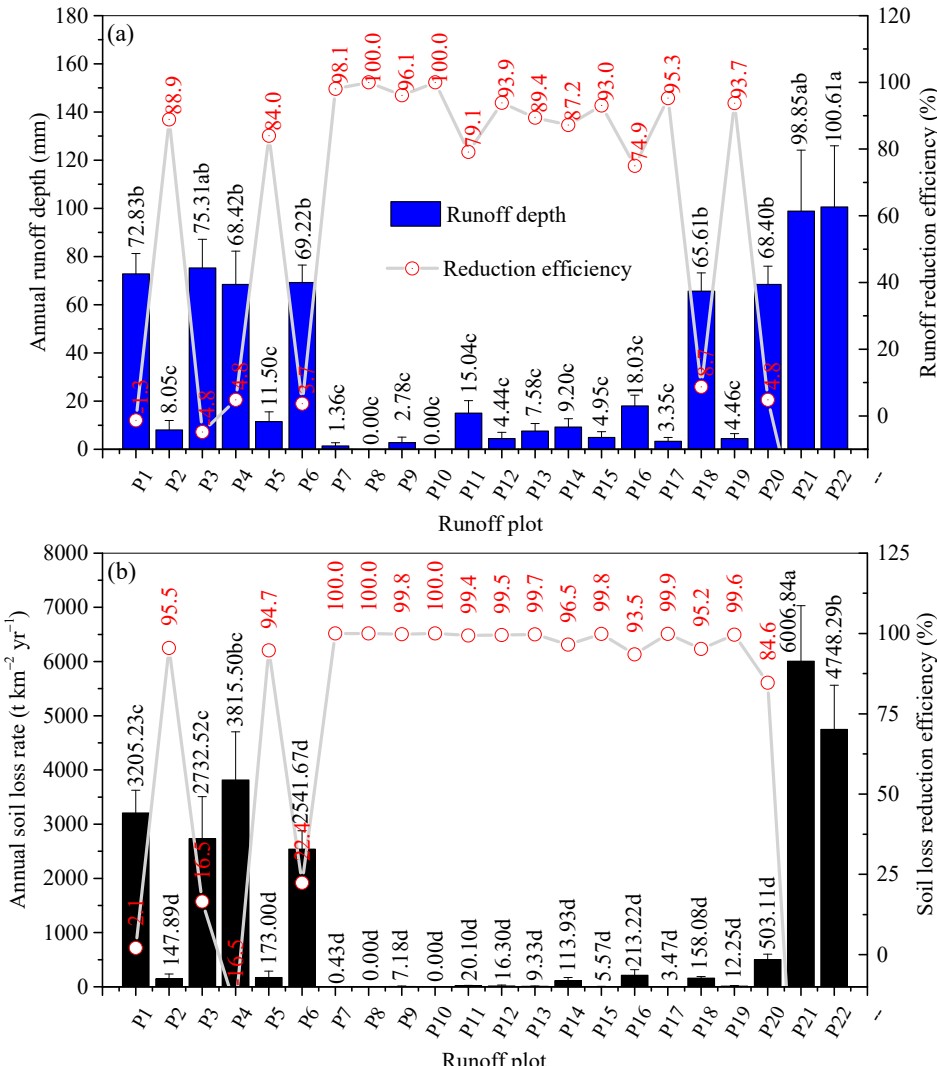

**Figure 4.** Annual runoff depth (**a**) and annual soil loss rate (**b**) and their reduction efficiencies by soil conservation measures in comparison to the mean values of bare plots 3 and 4 in 2014–2020 for the 22 runoff plots. Notes: the bars represent standard error. The average values on the columns with the same letter are not significantly different at *p* = 0.05, as determined by Fisher's protected least test.

**Table 3.** Characteristics of threshold rainfall, runoff, and soil loss characteristics for the 22 plots in 2014–2020.

| Plot | TRA (mm) * | MAP (mm) | Accu. P (mm) | RGT Times | Runoff Depth (mm) | | | | Soil Loss Rate (t km$^{-2}$ year$^{-1}$) | | | |
|---|---|---|---|---|---|---|---|---|---|---|---|---|
| | | | | | Min. | Max. | Mean | St.D | Min. | Max. | Mean | St.D |
| 1 | 4.8 | 280.39 | 1962.7 | 62 | 0.00 | 29.65 | 7.73b | 0.77 | 0.00 | 2278.23 | 339.95cd | 56.10 |
| 2 | 21.8 | 107.31 | 751.1 | 14 | 0.00 | 18.08 | 0.85c | 0.32 | 0.00 | 364.16 | 15.69e | 6.38 |
| 3 | 4.8 | 274.13 | 1918.9 | 60 | 0.00 | 18.08 | 7.99b | 0.75 | 0.00 | 2429.35 | 289.81d | 56.99 |
| 4 | 8.4 | 268.04 | 1876.3 | 57 | 0.00 | 18.08 | 7.26b | 0.74 | 0.00 | 2466.32 | 404.67c | 66.56 |
| 5 | 15.1 | 95.47 | 668.3 | 17 | 0.00 | 11.53 | 1.22c | 0.31 | 0.00 | 403.87 | 18.35e | 7.64 |
| 6 | 8.4 | 272.71 | 1909.0 | 59 | 0.00 | 18.08 | 7.34b | 0.72 | 0.00 | 2687.83 | 269.57d | 53.04 |
| 7 | 55.0 | 41.36 | 289.5 | 4 | 0.00 | 3.62 | 0.14c | 0.08 | 0.00 | 1.27 | 0.05e | 0.03 |
| 8 | 0 | 0.00 | 0.0 | 0 | 0.00 | 0.00 | 0.00c | 0.00 | 0.00 | 0.00 | 0.00e | 0.00 |
| 9 | 46.1 | 25.56 | 178.9 | 3 | 0.00 | 11.97 | 0.29c | 0.20 | 0.00 | 28.63 | 0.76e | 0.53 |
| 10 | 0 | 0.00 | 0.0 | 0 | 0.00 | 0.00 | 0.00c | 0.00 | 0.00 | 0.00 | 0.00e | 0.00 |
| 11 | 15.1 | 96.19 | 673.3 | 15 | 0.00 | 18.08 | 1.60c | 0.43 | 0.00 | 37.21 | 2.13e | 0.78 |
| 12 | 55.0 | 60.29 | 422.0 | 5 | 0.00 | 17.01 | 0.47c | 0.28 | 0.00 | 107.28 | 1.73e | 1.63 |
| 13 | 15.1 | 96.86 | 678.0 | 14 | 0.00 | 8.75 | 0.80c | 0.24 | 0.00 | 23.13 | 0.99e | 0.44 |
| 14 | 12.6 | 86.77 | 607.4 | 16 | 0.00 | 10.55 | 0.98c | 0.27 | 0.00 | 387.06 | 12.08e | 6.16 |
| 15 | 15.1 | 75.69 | 529.8 | 10 | 0.00 | 5.21 | 0.53c | 0.17 | 0.00 | 18.78 | 0.59e | 0.31 |
| 16 | 15.1 | 134.26 | 939.8 | 21 | 0.00 | 14.92 | 1.91c | 0.46 | 0.00 | 568.77 | 22.61e | 10.36 |
| 17 | 28.7 | 78.30 | 548.1 | 8 | 0.00 | 5.42 | 0.36c | 0.13 | 0.00 | 7.71 | 0.37e | 0.16 |
| 18 | 8.4 | 255.69 | 1789.8 | 53 | 0.00 | 18.08 | 6.96b | 0.76 | 0.00 | 130.40 | 16.77e | 3.09 |
| 19 | 15.1 | 75.97 | 531.8 | 9 | 0.00 | 9.22 | 0.47c | 0.18 | 0.00 | 73.65 | 1.30e | 1.12 |
| 20 | 8.4 | 264.46 | 1851.2 | 55 | 0.00 | 31.45 | 7.25 b | 0.80 | 0.00 | 920.36 | 53.36e | 16.54 |
| 21 | 4.8 | 280.36 | 1962.5 | 64 | 0.00 | 81.04 | 10.49ab | 1.74 | 0.00 | 2534.04 | 637.09a | 79.97 |
| 22 | 4.8 | 278.50 | 1949.5 | 63 | 0.00 | 88.23 | 10.67a | 1.83 | 0.00 | 3024.96 | 503.61b | 75.53 |

Notes: * indicates the recorded minimum value to generate runoff in the 66 rainfall events. TRA, MAP, Accu. P, and RGT represent the threshold of the runoff amount, mean annual rainfall amount, accumulated rainfall amount, and runoff generation times, respectively. The mean values of runoff depth and soil loss rate on the columns with the same letter were not significantly at the 0.05 level.

Annual SLR also varied from 0 on plots 8 and 10 to 6006.84 t km$^{-2}$ year$^{-1}$ on plot 21 (Figure 4b), and annual SLRs on the plots 1, 3, 4, 6, 21, and 22 were higher than 2500 t km$^{-2}$ year$^{-1}$, which is above twelve times the tolerate value (i.e., 200 t km$^{-2}$ year$^{-1}$) in the region [15]. In the Miyun Reservoir catchment, these lands are around 600 km$^2$, occupying around 5% of the catchment area. Although this percentage is smaller, it should also be paid attention to, because it is still an important pollutant source [13,14], implying that these kinds of lands should have soil conservation measures implemented. In contrast, the annual SLRs from plots 2, 5, and 7–19 were much lower. In terms of event SLRs, the maximum SLR was up to 637 t km$^{-2}$ event$^{-1}$ from plot 21, and plots 1, 3, 4, 6, and 22 also had high event SLRs. Conversely, other plots had a much lower event SLR than 25 t km$^{-2}$ event$^{-1}$, and plots 8 and 10 did not yield sediment (Table 3).

### 3.3. Impact of the Slope Gradient and Slope Length

The effect of the slope gradient on runoff and SLR is a controversial issue. As the slope gradients increase, surface runoff and soil loss can increase, decrease, and/or change little, resulting from the influence of complex factors, such as a forming or disruption of soil crust [27,28], rill and/or ephemeral gully development [29,30], differential soil cracking [31], and/or ponding depth [32,33], etc. Sometimes, a threshold maximum value of H or SLR exists with increasing slope gradients, resulting from the interaction of runoff generation and its sediment carrying capacity [28,34]. In the present study, both annual H and annual SLR increased first and then decreased, with a threshold annual H of 100.1 mm and a threshold annual SLR of 4748.29 t km$^{-2}$ year$^{-1}$ occurring at the 14.4° slope (Figure 5). A number of studies [35–38] found that the threshold slope gradients of H and SLR varied from 25° to 50°, which are much steeper than the one used in this study. The threshold slope gradient of 14.4° of annual H and annual SLR can be explained by the impact of soil conservation measures. In the present study, the bare plots with a slope degree of 14.4° had the highest annual H and annual SLR, whereas other plots with slope degrees lower

and/or higher than 14.4° were implemented with soil conservation measures, and yielded lower annual Hs and annual SLRs. For the plots with the same treatment, the changes of Hs and SLRs with the slope gradient varied depending on plot treatments. For example, higher annual Hs and SLRs occurred on contour tillage plot 6 than those of contour tillage plots 18 and 20. Similar phenomenon also occurred for the terraced orchard plots 2 and 7 (Figure 4). This phenomenon could be attributed to the decrease in depressional storage and ponding depth on the plots with higher slope degrees [33,39]. However, for the shrub plots (i.e., plots 5, 9, 11, 13, 14, and 19), a threshold slope seemed to exist, whereas Hs and SLRs on plots 21 and 22 with a lower slope gradient were much higher than that of plot 4, resulting from the effect of the slope length.

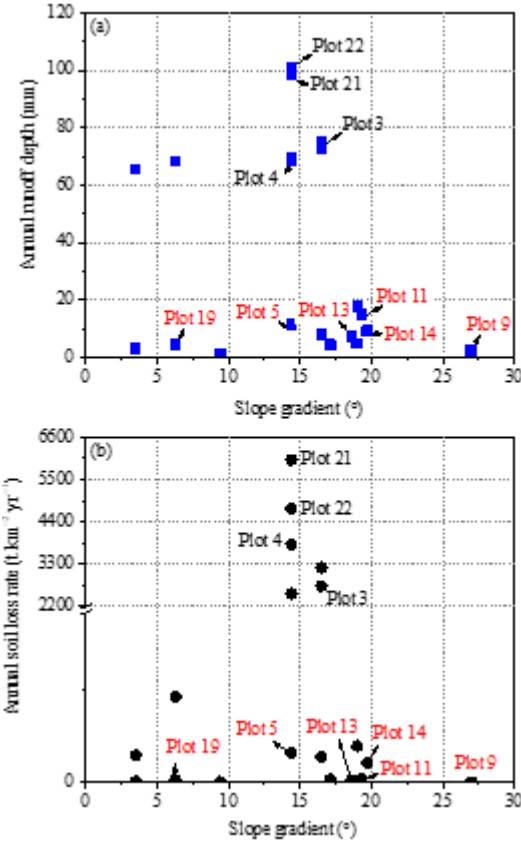

**Figure 5.** Change patterns of annual runoff depth (**a**) and annual soil loss rate (**b**) with increasing slope degree in 2014–2020 for the 22 runoff plots. Note: the plots in black letters represent bare plots, and those in red letters present shrub plots in Table 1.

Runoff and soil loss also present different changing patterns with increasing slope length [40–42]. In the present study, only three slope lengths (i.e., 5 m, 10 m, and 20 m) were included (Table 1). The scattered dots in Figure 6 illustrate that annual Hs and SLRs increased or decreased when the slope lengths increased from 5 m to 20 m, resulting from the impact of soil conservation measures. Further analysis was done when their impacts were excluded. The annual H decreased when the slope length increased from 5 m to 10 m for plots 11 and 14, which had similar coverage rates and slope degrees (Table 1). This is because the time and volume of rainfall per unit area required before the runoff starts are higher in the longer plot than the smaller one [43]. However, for the bare plots 3, 4, 21, and 22, the annual H increased with increasing slope length. The difference could result from different earth surface conditions. According to study results by Mermut et al. [44] and Fang et al. [28], the soil texture in the study region made the soil crust easily form on the bare plot, which reduced the infiltration rate and increased the runoff generation capacity [33]. Although soil crust was not detected in the present

study, it has been verified by previous studies through the use of natural rainfall [33] and rainfall simulation experiments [28]. Given the same land surface, annual SLRs increased with increasing slope length. For example, the mean annual SLRs of bare plots 21 and 22 were higher than those of bare plots 3 and 4, resulting from an increased surface runoff amount with increasing slope length. In contrast, a higher SLR on brush plot 14 than that on brush plot 11 could be explained by increased sediment concentration in runoff due to the increased concentrated flow on a longer plot [28].

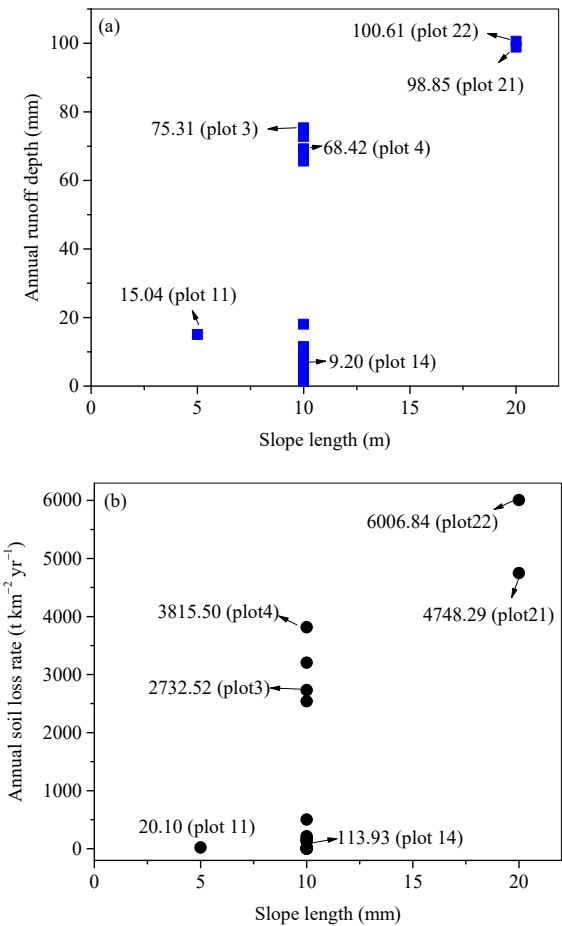

**Figure 6.** Annual runoff depth (**a**) and soil loss rate (**b**) for the 22 runoff plots in 2014–2020 with slope length.

*3.4. Impact of Rainfall Characteristics*

Rainfall is the external force inducing surface runoff and soil loss. In the present study, five rainfall eigenvalues were used to explore the impact of rainfall characteristics on event H and SLR. Pearson correlation coefficients indicated that event H was significantly correlated with event P and rainfall intensity ($I_{30}$ and/or $I_{60}$) at the 0.01 level, excluding the terraced plots 2 and 7, and the densely vegetated plot 12 (Table 4), the event H and SLR of which, however, were significantly correlated with P and RD. In respect to event SLR, event SLR was significantly correlated with P, $I_{30}$, and/or both of them on most of the plots. Different responses of H and SLR from the plots to different rainfall eigenvalues can result from the protection capacities of the measures [45–47]. For instance, the vegetated measures can effectively protect the splash impact of rainfall intensity, resulting in insignificant correlations of H and rainfall intensity for the vegetated plots.

**Table 4.** Pearson coefficient matrix between the five rainfall eigenvalues, event runoff depth (H), and event soil loss rate SLR for 66 rainfall events in 2014–2020.

|  |  | RD | P | $I_m$ | $I_{30}$ | $I_{60}$ |
|---|---|---|---|---|---|---|
| H | P1 | 0.293 * | 0.642 ** | 0.058 | 0.546 ** | 0.407 ** |
|  | P2 | 0.344 ** | 0.412 ** | −0.117 | 0.212 | 0.146 |
|  | P3 | 0.379 ** | 0.601 ** | 0.008 | 0.473 ** | 0.368 ** |
|  | P4 | 0.300 * | 0.544 ** | 0.040 | 0.477 ** | 0.358 ** |
|  | P5 | 0.102 | 0.306 * | 0.179 | 0.410 ** | 0.308 * |
|  | P6 | 0.263 * | 0.566 ** | 0.117 | 0.651 ** | 0.460 ** |
|  | P7 | 0.522 ** | 0.524 ** | −0.121 | 0.206 | 0.131 |
|  | P9 | −0.100 * | 0.168 | 0.209 | 0.166 | 0.300 * |
|  | P11 | 0.069 | 0.300 * | 0.211 | 0.395 ** | 0.293 * |
|  | P12 | 0.248 * | 0.574 ** | −0.020 | 0.122 | 0.148 |
|  | P13 | 0.070 | 0.260 * | 0.198 | 0.398 ** | 0.259 * |
|  | P14 | −0.007 | 0.257 * | 0.254 * | 0.409 ** | 0.321 ** |
|  | P15 | 0.196 | 0.359 ** | 0.195 | 0.365 ** | 0.250 * |
|  | P16 | 0.181 | 0.537 ** | 0.171 | 0.514 ** | 0.410 ** |
|  | P17 | 0.399 ** | 0.656 ** | −0.019 | 0.330 ** | 0.294 * |
|  | P18 | 0.385 ** | 0.642 ** | 0.041 | 0.520 ** | 0.369 ** |
|  | P19 | 0.269 * | 0.366 ** | −0.025 | 0.131 | 0.106 |
|  | P20 | 0.281 * | 0.643 ** | 0.073 | 0.553 ** | 0.410 ** |
|  | P21 | 0.555 ** | 0.640 ** | −0.057 | 0.394 ** | 0.274 * |
|  | P22 | 0.567 ** | 0.682 ** | −0.039 | 0.403 ** | 0.318 ** |
| SLR | P1 | 0.002 | 0.310 * | 0.188 | 0.431 ** | 0.317 ** |
|  | P2 | 0.051 | 0.229 | 0.089 | 0.283 * | 0.151 |
|  | P3 | −0.014 | 0.298 * | 0.223 | 0.410 ** | 0.366 ** |
|  | P4 | −0.110 | 0.265 * | 0.290 * | 0.436 ** | 0.388 ** |
|  | P5 | −0.100 | 0.099 | 0.176 | 0.207 | 0.167 |
|  | P6 | 0.082 | 0.347 ** | 0.168 | 0.441 ** | 0.335 ** |
|  | P7 | 0.337 ** | 0.408 ** | −0.088 | 0.126 | 0.023 |
|  | P9 | −0.129 | 0.055 | 0.221 | 0.082 | 0.245 * |
|  | P11 | −0.076 | 0.221 | 0.129 | 0.128 | 0.170 |
|  | P12 | 0.460 ** | 0.420 ** | −0.077 | 0.265 * | 0.240 |
|  | P13 | −0.154 | 0.000 | 0.210 | 0.319 ** | 0.155 |
|  | P14 | −0.024 | 0.201 | 0.070 | 0.199 | 0.181 |
|  | P15 | −0.096 | 0.024 | 0.225 | 0.204 | 0.097 |
|  | P16 | −0.052 | 0.306 * | 0.113 | 0.179 | 0.225 |
|  | P17 | 0.004 | 0.359 ** | 0.126 | 0.218 | 0.304 * |
|  | P18 | −0.109 | 0.273 * | 0.172 | 0.290 * | 0.246 |
|  | P19 | 0.027 | −0.013 | −0.064 | −0.048 | −0.064 |
|  | P20 | 0.295 * | 0.498 ** | 0.079 | 0.396 ** | 0.373 ** |
|  | P21 | 0.035 | 0.334 ** | 0.239 | 0.525 ** | 0.408 ** |
|  | P22 | 0.004 | 0.315 ** | 0.284 * | 0.519 ** | 0.463 ** |

Note: * represents significance at the 0.05 level, and ** represents significance at the 0.01 level. $P$, $RD$, $I_m$, $I_{30}$, and $I_{60}$ represent rainfall amount, rainfall duration, mean rainfall intensity, and maximum 30- and 60-min rainfall intensity for each erosive rainfall event.

Similar to the studies by Fang et al. [33] and Zhu and Zhu [7], a large percent of annual runoff and soil loss occurred during several rainfall events. In the present study, we compared soil loss by the maximum annual erosive event to annual soil loss through four exampled plots, which represented different soil conditions (i.e., plots 1–3 and 14) in Figure 7. Soil loss from cultivated plot 1 by the maximum annual soil erosive event in 2014–2020 ranged from 1658.67 to 5104.98 t km$^{-2}$, which contributed to a mean value of 40.50% of the annual total soil loss, ranging from 30.72% in 2019 to 61.13% in 2018. The magnitude and contributions of the maximum annual erosive event on bare plot 3 were similar to those of plot 1. However, much higher contributions of the maximum annual erosive event occurred on plots 2 and 14, with terrace and shrub measures, respectively. The maximum contributions of the maximum annual erosive events were up to 85% on

plot 2 and 89% on plot 14, although their maximum soil loss rates were much less than those on the bare and cultivated plots.

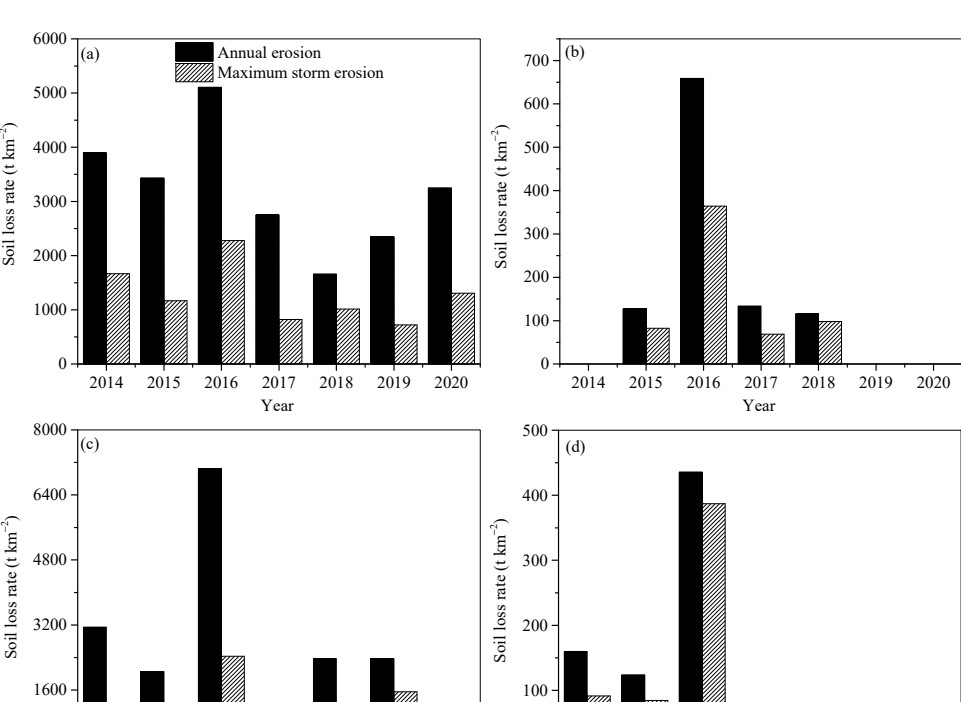

**Figure 7.** Soil loss in the maximum annual erosive event and annual soil loss on four types of land use and soil conservation measures: plot 1 (**a**), plot 2 (**b**), plot 3 (**c**), and plot 14 (**d**). The information of the plots was given in Table 1.

### 3.5. Impact of Soil Conservation Measures

In order to study the impact of soil conservation measures, ten kinds of land use and soil conservation measures were obtained by using the 22 plots in Table 1 (Figure 8). Among the ten treatments, mean annual Hs differed greatly, and changed from 85.80 mm on bare plots, 72.83 mm on the cultivated plot without any measure, 67.74 mm on the cultivated plots with contour tillage, 11.49 mm on the grass plots, followed by the shrub plots, orchard plots with terrace, and other plots. Noticeably, no runoff occurred on plot 8 with forest plus grass measures and plot 10 with forest and fish scale pit measures. In comparison to the bare plots, the mean runoff reduction efficiencies were above 86% for the plots with soil conservation measures, excluding plots 1 and 6. Based on this result, the large-scale implementation of soil conservation measures since 2000 in the study region could, to some extent, explain the decreased discharge into the Miyun Reservoir, which agrees with the former studies at a catchment scale [13–15]. In order to ensure water quantity of the Miyun Reservoir, soil conservation measures should be water-saving, which allows more runoff downstream to the reservoir.

Annual SLRs presented the same changing patterns. The bare plots had the largest SLR, with an average of 4325.79 t km$^{-2}$ year$^{-1}$, followed by the cultivated plot with no soil conservation measures, with an average of 3205.23 t km$^{-2}$ year$^{-1}$, and the cultivated plot with contour tillage, with an average of 1067.62 t km$^{-2}$ year$^{-1}$. In contrast, other kinds of plots had much lower mean annual SLRs. The mean annual SLR was 109.39 t km$^{-2}$ year$^{-1}$ for the grass plots, 74.16 t km$^{-2}$ year$^{-1}$ for the orchard plot with terrace, 55.97 t km$^{-2}$ year$^{-1}$ for the shrub plots, and 16.3 t km$^{-2}$ year$^{-1}$ for the forest plus shrub plots. No sediment ran out of the plots with forest plus fish scale pit and/or grass measures. Noticeably, almost all of the plots with soil conservation measures had higher soil loss reduction efficiencies

of above 95%. The effective reductions of soil and water losses by vegetated measures were consistent with a number of studies [47–51]. Consistent with previous studies [26,47], terrace was effective in reducing runoff and soil loss. In the present study, soil loss from cultivated plot 17 with terrace was nearly 0 (i.e., 3.47 t km$^{-2}$ year$^{-1}$), and no runoff and soil loss occurred on plot 10, which had terrace and forest measures implemented, although its vegetation coverage was not very high. This implies that the combination of vegetation and terrace measures was more effective than a single measure in reducing soil and water losses [8]. In the present study, the mean annual SLR on plot 6 with a slope degree of 14.4° was up to 2541 t km$^{-2}$ year$^{-1}$, which is twelve times above the tolerable rate, although contour tillage was also implemented on this plot (Figure 3). In contrast, contour tillage can effectively reduce soil loss from plots 18 and 20, which have lower slope degrees (Table 1). This implies that contour tillage is not reasonable when it is implemented on steep slopes in the study region.

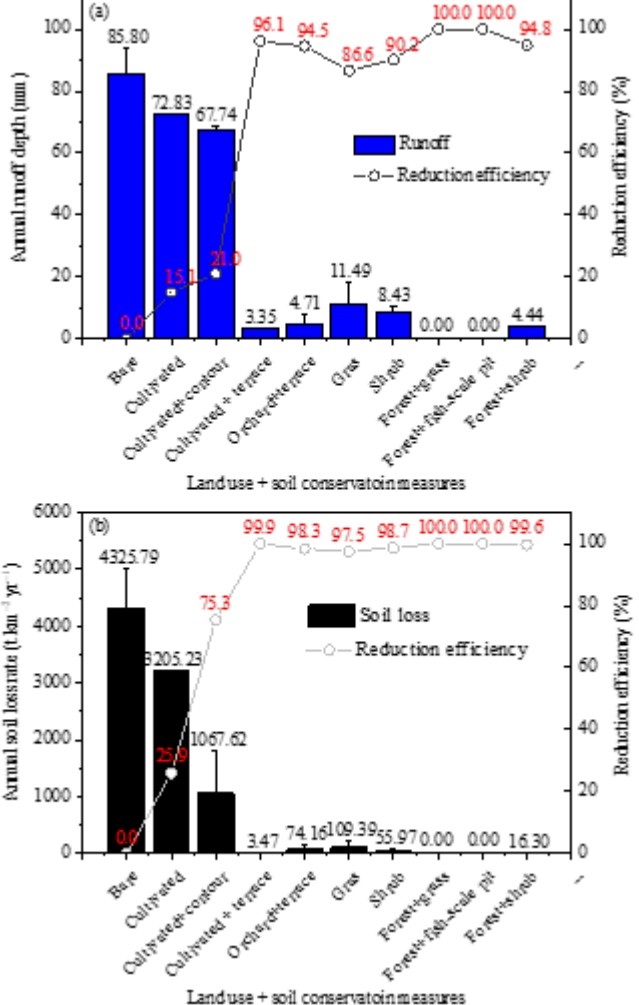

**Figure 8.** Annual runoff depth (**a**) and soil loss rate (**b**) under different land use and soi conservation measures for the study plots in 2014–2020. Note: On the x-axis, bare represents plots 3, 4, 21, and 22, cultivated represents plot 1, cultivated + contour represents plots 6, 18, and 20, and cultivated + terrace represents plot 17. Orchard represents plot 2 and 7, shrub represents plots 5, 9, 11, 13, 14, and 19, forest + grass represents plot 8, forest + fish-scale pit represents plot 10, forest + shrub represents plot 12, and grass represents plots 15 and 16 in Table 1.

The implemented soil conservation measures also significantly interfered with the relationships between event H and event SLR. Table 5 demonstrated that, except for the bare and cultivated plots, the SLR cannot be fitted at the 0.01 level.

**Table 5.** Relationships between event runoff depth (H) and soil loss rate (SLR) (SLR = $aH^b$) for the 22 runoff plots.

| Plot | N | a | b | F-Test | $R^2$ | Sig. |
|------|----|--------|--------|--------|-------|-------|
| 1 | 62 | 7.004 | 1.598 | 57.489 | 0.489 | 0.000 |
| 2 | 14 | 19.723 | 0.309 | 0.178 | 0.015 | 0.680 |
| 3 | 60 | 7.742 | 1.300 | 28.874 | 0.300 | 0.000 |
| 4 | 57 | 13.710 | 1.310 | 42.928 | 0.434 | 0.000 |
| 5 | 17 | 1.173 | 2.056 | 7.371 | 0.329 | 0.016 |
| 6 | 59 | 3.579 | 1.804 | 87.412 | 0.605 | 0.000 |
| 7 | 4 | 0.279 | 0.804 | 0.240 | 0.107 | 0.672 |
| 9 | 3 | 1.385 | 1.140 | 0.313 | 0.238 | 0.675 |
| 11 | 15 | 5.900 | −0.087 | 0.019 | 0.001 | 0.894 |
| 12 | 5 | 0.415 | 1.218 | 0.729 | 0.195 | 0.456 |
| 13 | 14 | 0.685 | 1.031 | 7.104 | 0.372 | 0.021 |
| 14 | 16 | 3.318 | 1.281 | 4.100 | 0.227 | 0.062 |
| 15 | 10 | 0.296 | 1.625 | 4.711 | 0.371 | 0.062 |
| 16 | 21 | 1.937 | 1.227 | 3.959 | 0.180 | 0.062 |
| 17 | 8 | 3.698 | −0.500 | 0.521 | 0.08 | 0.497 |
| 18 | 53 | 1.958 | 0.842 | 14.988 | 0.227 | 0.000 |
| 19 | 9 | 0.312 | 1.181 | 0.681 | 0.089 | 0.436 |
| 20 | 55 | 2.650 | 0.949 | 12.868 | 0.195 | 0.001 |
| 21 | 64 | 56.082 | 0.905 | 34.756 | 0.359 | 0.000 |
| 22 | 63 | 22.189 | 1.203 | 60.437 | 0.502 | 0.000 |

Note: the zero values for the 66 erosive events were assigned to 0.01 for power function regression. The relationships between the H and SLR were not given for plots 8 and 10 because no runoff occurred on these two plots.

## 4. Conclusions

The latest seven years (2014–2020) of monitored data from 22 runoff plots in the upstream catchment of the Miyun Reservoir in Beijing, Northern China were employed to study soil and water loss characteristics on slopes and their responses to slope, rainfall, and soil conservation measures, and some findings and implications were obtained. The bare land suffered the severest soil loss, with a mean annual SLR of 4325 t km$^{-2}$ year$^{-1}$. Cultivated lands without any measures also experienced severe soil loss, with an annual SLR above 3205 t km$^{-2}$ year$^{-1}$. Contour tillage cannot effectively control soil loss on the cultivated land with steep slopes. Soil conservation measures, such as forest, shrub, grass, terrace, and fish scale pits, as well as their combinations, can effectively reduce soil and water losses, with soil and water loss reduction efficiencies higher than 95% and 86%, respectively.

To protect drinking water safety, water quantity is a priority in the study region, and water-saving soil control measures should be considered to resolve the contradiction of soil loss control and water shortage in the downstream reservoir. The vegetation coverage of forest, shrub, and grass should be sparser, which allows more runoff to run downslope and simultaneously and effectively control soil loss. The size of the terrace on gentle slopes should be carefully designed. Bare and cultivated lands suffered the most severe soil erosion, with SLRs above 2732 t km$^{-2}$ year$^{-1}$, and soil conservation measures should be implemented. Cultivated land with contour tillage still suffered a very high SLR (i.e., 2541 t km$^{-2}$ year$^{-1}$), and terrace or other vegetation measures (e.g., grass strips) should be implemented on this kind of land, because these types of measures can greatly reduce SLRs in the study region (Figure 4b).

Future studies can be conducted using physically based soil erosion models, such as the Water Erosion Prediction Plan (WEPP) and Soil and Water Assessment Tool (SWAT), to

evaluate the effect of soil conservation measures on different slopes on downstream water resources.

**Funding:** This work was financially funded by the Beijing Natural Science Foundation (grant number 8202045) and the National Natural Science Foundation of China (grant number 41977066).

**Institutional Review Board Statement:** Not applicable.

**Informed Consent Statement:** Not applicable.

**Data Availability Statement:** The data presented in the present study are available on request from the corresponding author.

**Acknowledgments:** Special thanks are owed to the two reviewers for their invaluable suggestions to the improvement of the manuscript.

**Conflicts of Interest:** The author declares no conflict of interest.

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
