# Peer review of "Water-Saving Soil Conservation Measures Should Be Used in Northern China: Evidence from Runoff Plot Data"

_water, doi:10.3390/w13060853_

Round 1
Reviewer 1 Report
First of all, I want to congratulate the authors for their efforts. The authors presented a lon-term study about the erosion in 22 different plots. This si an interesting and updated issue. The topic is close to the journal scope. Although their results have a high impact, the paper have some problems, which must be solved in order to enhance the quality of their manuscript. Following I include my suggestions:
At the beginning of the abstract, the authors have to add a sentence or two defining the problem that they address.
All the acronyms must be defined the first time that they are used. Check SLR in the abstract.
The last part of the abstract (from “Water-saving measures should be” till the end) must be summarized. Authors use too much space in the abstract for conclusions/recommendations. They have to highlight their results and define in more detail their methodology.
In the keywords authors have to avoid using the same words than in the title.
The paragraph with the aim of the paper must be extended and a new paragraph detailing the structure of the paper should be added after the aim of the paper.
In Figure 1 the DEM of the area with the plots should be included. Furthermore, Figure 1 must be cited in the text and explained in detail.
Authors have to unify the nomenclature, in some cases the use “m” and in other “meters”. Please crosscheck the paper and correct these issues.
In all the paper, including tables and figures, the scientific names of sp must be in italics. Please check this issue.
Obtained models presented in Figure 5 must be explained in detail. If te model included in Figure 5 is obtained form the data of the Figures it should be modified. In the current figures it is hard to analyze the models since plots with different characteristic (different measures for erosion attenuation) are compared. It is not possible to compare the effect of slope having such diversity of data. It should be better to focus and use only the plots without any measure to stop the erosion. Same comment for Figure 6. It is not possible to compare data having this diversity. Authors should focus and compare plots with similar characteristics (in terms of measures to minimize the erosion).
At the end of conclusions authors must include a short paragraph detailing the future work.
Author Response
Dear reviewer,
Many thanks for your work. According to the suggestions or comments, careful revisions have been done for each comment. All the revised sections or words are labeled in the tracked manuscript.
1, At the beginning of the abstract, the authors have to add a sentence or two defining the problem that they address.
Response: Yes, one sentences was added in the revised manuscript.
2, All the acronyms must be defined the first time that they are used. Check SLR in the abstract.
Response: Thanks. It was done.
3, The last part of the abstract (from “Water-saving measures should be” till the end) must be summarized. Authors use too much space in the abstract for conclusions/recommendations. They have to highlight their results and define in more detail their methodology.
Response: Yes. In order to highlight the results, this suggested part was summarized in the revised manuscript.
4, In the keywords authors have to avoid using the same words than in the title.
Response: Thanks. The two phrases that were in the title were deleted from the keywords
5, The paragraph with the aim of the paper must be extended and a new paragraph detailing the structure of the paper should be added after the aim of the paper.
Response: Thanks for your suggestion. According to the suggestion, the study aims were extended, and the structure of this manuscript was also given as a paragraph. Specific revision contents were at the end of the introduction.
6, In Figure 1 the DEM of the area with the plots should be included. Furthermore, Figure 1 must be cited in the text and explained in detail.
Response: The current DEM I have is only 30-m resolution. However, the width of the plots is only 5-m, and the lengths of plots are less than 20-m which means that the runoff plots cannot be better displayed on the DEM. This figure was cited in the text and explained in the revised manuscript.
7, Authors have to unify the nomenclature, in some cases the use “m” and in other “meters”. Please crosscheck the paper and correct these issues.
Response: Thanks. They are unified in the revised manuscript.
8, In all the paper, including tables and figures, the scientific names of sp must be in italics. Please check this issue.
Response: Thanks. What is the mean of “sp”? Is it Ps? Similarly, other abbreviations such as RD, Im, I30, and I60 were also be in italics in the revised manuscript.
9, Obtained models presented in Figure 5 must be explained in detail. If the model included in Figure 5 is obtained from the data of the Figures it should be modified. In the current figures it is hard to analyze the models since plots with different characteristic (different measures for erosion attenuation) are compared. It is not possible to compare the effect of slope having such diversity of data. It should be better to focus and use only the plots without any measure to stop the erosion. Same comment for Figure 6. It is not possible to compare data having this diversity. Authors should focus and compare plots with similar characteristics (in terms of measures to minimize the erosion).
Response: The distribution of the dots in Figure 4 indicated that there existed a threshold gradient or slope length, and a curve was plotted. However, there are a variety of plots with different soil conservation measures. Yes, as suggested as the reviewer, it is better to compare runoff depths and soil loss rates with the same land use. Whereas only bare and shrub plots had over 4 plots, each kind of other plots were not more than three plots (Table 1). Therefore, the curves were deleted and only the bare and shrub plots were labeled in the plots to investigate the effect of slope gradient on runoff and soil loss. Correspondingly, some discussions were edited and given in the text. This may be a good way to discuss the effect of slope gradient or length on runoff and soil loss. Therefore, Figure 6 was kept in its original form.
10, At the end of conclusions authors must include a short paragraph detailing the future work.
Response: Yes. One short paragraph was added at the end of the revised manuscript.
Reviewer 2 Report
Comments to Authors
The research paper presents an interesting issue related to the importance of Water-saving conservation measures. However I notice, the paper addresses a topic that is abundant in the literature that compromises the novelty. However, there are some aspects that should be revised in order to improve paper quality.
The introduction is well written however the novelty of the study is not reported. I suggest revising the last part of the introduction because it should report the novelty of the work and how the study can affect local communities and stakeholders. Moreover, a working hypothesis and what did you aspect from the evidence of plot data
The 2.2 part of the methodology explains in detail the plot design. You should provide an indication about the insertion depth of boundaries. I think 30-cm deep insertion should be ensured to avoid subsurface runoff entrance/exit.
For a better comparison among plots for each land use, one no-conservation measure should be considered.
Results are well arranged and discussion seems targeted
Please, report in the results how many events were accounted
In figure 3 use the same scale for the y-axes
For runoff, soil loss and rainfall, at event scale (in addition to annual scale) a chart or a table should be provided
On page 8 when you write about the tolerable value of soil loss, you should provide a reference.
Please, clarify the choice of the 4 example plot. I see that they are very different but a comparison is impossible. I suggest expanding the sample by choosing couples of plots (e.g. cultiv. vs. cultiv. with conserv. practices; bare soil vs bare soil with cons pract. etc.)
Author Response
Dear reviewer,
Many thanks for your work. According to the suggestions or comments, careful revisions have been done for each comment. All the revised sections or words are labeled in the tracked manuscript.
1, The introduction is well written however the novelty of the study is not reported. I suggest revising the last part of the introduction because it should report the novelty of the work and how the study can affect local communities and stakeholders. Moreover, a working hypothesis and what did you aspect from the evidence of plot data
Response: Great thanks for your suggestion. The novelty and a hypothesis were added to the third to the last paragraph.
2, The 2.2 part of the methodology explains in detail the plot design. You should provide an indication about the insertion depth of boundaries. I think 30-cm deep insertion should be ensured to avoid subsurface runoff entrance/exit.
Response: Yes. The height of the boundary is around 20 cm above the ground, and around 30-cm depth underground to prevent runoff from entering the ground. This information was added in the revised manuscript.
3, For a better comparison among plots for each land use, one no-conservation measure should be considered.
Response: Yes, the plots with no soil conservation measures were used as a reference to compare the effect of soil conservation measures on runoff and soil loss.
4, Results are well arranged and discussion seems targeted. Please, report in the results how many events were accounted.
Response: There were 66 erosive rainfall event, this information was at the second paragraph of 3.1 section.
5, In figure 3 use the same scale for the y-axes
Response: The Y-scales of each figure in Figure 3 were unified.
6, For runoff, soil loss and rainfall, at event scale (in addition to annual scale) a chart or a table should be provided
Response: Thanks. Yes, more data was shown in the revised Table 3 that gave runoff, soil loss and rainfall data at an event scale.
7, On page 8 when you write about the tolerable value of soil loss, you should provide a reference.
Response: Yes, a reference was added after the sentence that contained the tolerable value of soil loss in the study area.
8, Please, clarify the choice of the 4 example plot. I see that they are very different but a comparison is impossible. I suggest expanding the sample by choosing couples of plots (e.g. cultiv. vs. cultiv. with conserv. Practices; bare soil vs bare soil with cons pract. etc.).
Response: The main aim of Figure 7 is to show the impact of single rainfall event on annual total soil loss. The four plots represent different soil surfaces (i.e., cultivated land, bare land, orchard with terrace, and vegetation covered land). Of cource, the information that more soil loss contribution by one single rainfall event was also given for the plots with soil conservation measures. In order to make readers understand the aims, one sentence (which represented different soil conditions) was given in the revised manuscript.
Reviewer 3 Report
Attached file for minor corrections/edits. Lines 58-61 should include new text that summarizes results from the cited references. I saw no text references to figures 1 and 8.

Author Response
Dear reviewer,
Many thanks for your work. According to the suggestions in the word and pdf file, careful revisions have been done for each comment. All the revised sections or words are labeled in the tracked manuscript.
1, Attached file for minor corrections/edits. Lines 58-61 should include new text that summarizes results from the cited references. I saw no text references to figures 1 and 8.
Response: Thanks, summarized results were given in lines 58-61 in the original manuscript.
Figures 1 and 8 were also cited in the revised manuscript.
2, the comments relating to small grammar problems in the pdf file
Response: They are corrected one by one in the revised manuscript.
3, For figure 5, F is F-statistics in SPSS which was given in the revised manuscript. The numbers of the figures were three when they were given in SPSS software. Therefore, they were not changes.
Reviewer 4 Report
Specific comments on the manuscript
Abstract
L10 Substitute plot with plots.
L12 Use water-scarce instead of water-priority.
L16 Instead of bare land consider placing bare soil or uncultivated land.
L16 Instead of severest consider most severe.
L16 Consider first defining the abbreviation SLR, then make use of it in the Abstract.
L21 Substitute allows with allow.
L22 Instead of designed consider redesigned.
L23-L24 The sentence: “should be replaced by other types of measures or more measures should be added on this kind of land” should be rewritten into a scientifically sound conclusion.
Introduction
L36-37 Consider substituting the sentence “many studies have been done” by “have been published until very recently”.
L40 Elaborate on “the background levels” it is rather vague.
L45 I suggest you elaborate on the measures with a few sentences/words to strengthen your case.
L46-47 Provide examples of the measures you are referring in these two sentences.
L55-56 I suggest you move the questions posed and adjust them to the end of the Introduction section.
L57 Consider substituting the sentence “many studies have been done” by “many studies have taken place” or “have been performed”.
L65 Consider substituting the sentence “well been done in the world” by “have thoroughly been examined”.
Materials and methods
2.1
L80 No need for “in elevation".
L83 Instead of falling, occurring.
2.2
L95 10 m long.
L99 Instead of “was given”, “is provided”.
L107 Provide a reference for the dry method you used. If there is no reference available, describe the method and analysis you performed in brief.
2.2
L122 Instead of significantly use significant.
3.1
L141 Consider using hours and minutes instead of hour decimals.
L143 I strongly recommend you compare some of this rainfall durations with mean observed historical values, representative of the region. Also to provide some estimates of the future rainfall conditions, based on findings on climate change for the region.
3.2
L151-152 The sentence “Mean annual Hs were less than 503.11 mm on most of the plots, which were significantly less than those on the bare and cultivated plots 1, 3, 4, 6, 21, and 22” makes no sense, consider rewriting with a better explanation.
L167 “that these kinds of lands” how much of the region in hectares or km2 occupy these lands? I suggest you to provide a short analysis on the land use / land cover of the region, especially in relation to the “lands” of interest.
L170 high or higher?
3.3
L183 Consider substituting “which are much larger than the one in this study” by “characterize soils steeper that the ones used in this study”.
L208 Consider substituting “made soil crust easily forms” by “resulted in forming soil crusts on the bare plots' surface”.
3.5
L261 based on or upon
L263 this needs to be verified also with the amount of rainfall in catchment scale. If the rainfall amount the basin receives has decreased from 2000, then rainfall - and not the adopted measures - is the reason of reduced downstream discharge. Check the historical rainfall amounts to be sure of this suggestion. If verified, then use the finding. Now, it looks rather as an arbitrary conclusion.
L281 Correct to “In consistence”
L294 Consider using another verb instead of “destroyed”
5 Conclusions
The fact that the study presents a number of valuable measurements and data analysis, makes documenting an elaborate "conclusions" section, very important. This section needs to be rewritten to support much of the hypothesis.
L317 I recommend you provide here the findings of your study that support this hypothesis.
L319 I recommend you suggest measures to replace the contour tillage and provide evidence of your study to support this hypothesis here.
Table 1
Specify whether all the treatments were applied for all six years.
Use Italics in latin plant names (e.g. Vitex negundo).
Specify in the manuscript the method you used to calculate vegetation cover percentage in the plots.
Table 2
Correct to amount, not amouint
Consider changing the sentence “Percent of mean annual erosive rainfall amouint to the mean annual one” to “Percent of mean annual erosive rainfall amount in relation to the mean annual”
Table 3
I recommend you provide all the available information in Table 3, such as 1. the cumulative annual, 2. the mean annual, 3. the event maximum and minimum both for water depth and SLR. This will collect the data in one place and assist the reader to better cope with the provided analysis in the manuscript.
Figure 3 caption
Provide the information that rainfall incidents data were recorded between 2014 - 2020.
Figure 6b caption
It is m, not mm
Author Response
Dear reviewer,
Many thanks for your work. According to the suggestions or comments, careful revisions have been done for each comment. All the revised sections or words are labeled in the tracked manuscript.
0, In the Abstract section
0.1, L10 Substitute plot with plots.
Response: It was corrected.
0.2, L12 Use water-scarce instead of water-priority.
Response: It was corrected.
0.3, L16 Instead of bare land consider placing bare soil or uncultivated land.
Response: It was done.
0.4, L16 Instead of severest consider most severe.
Response: It was done.
0.5, L16 Consider first defining the abbreviation SLR, then make use of it in the Abstract.
Response: It was done.
0.6, L21 Substitute allows with allow.
Response: This word was deleted because lines 20- 25 in the original manuscript were edited.
0.7, L22 Instead of designed consider redesigned.
Response: This word was deleted because lines 20- 25 in the original manuscript were edited.
0.8, L23-L24 The sentence: “should be replaced by other types of measures or more measures should be added on this kind of land” should be rewritten into a scientifically sound conclusion.
Response: This sentence was rewritten.
1, Introduction
1.1, L36-37 Consider substituting the sentence “many studies have been done” by “have been published until very recently”.
Response: Yes. The phrase has been edited.
1.2, L40 Elaborate on “the background levels” it is rather vague.
Response: Yes, a value was given after the phrase.
1.3, L45 I suggest you elaborate on the measures with a few sentences/words to strengthen your case.
Response: Yes. Some measures were given as “…such as contour tillage, no tillage, horizontal terracing, level ditching etc on…”
1.4, L46-47 Provide examples of the measures you are referring in these two sentences.
Response: Some examples of soil conservation measures were given in the revised manuscript.
1.5, L55-56 I suggest you move the questions posed and adjust them to the end of the Introduction section.
Response: Thanks. The sentence has been removed to the near end of the revised manuscript.
1.6, L57 Consider substituting the sentence “many studies have been done” by “many studies have taken place” or “have been performed”.
Response: Thanks. It was done.
1.7, L65 Consider substituting the sentence “well been done in the world” by “have thoroughly been examined”.
Response: This comment has also been commented. It was replaced by “have thoroughly been documented”.
- Materials and methods
2.1 L80 No need for “in elevation". L83 Instead of falling, occurring.
Response: Thanks. They were deleted or corrected.
2.2
L95 10 m long.
Response: Yes. It is done.
L99 Instead of “was given”, “is provided”.
Response: Yes. It was corrected.
L107 Provide a reference for the dry method you used. If there is no reference available, describe the method and analysis you performed in brief.
Response: A reference was cited in the revised manuscript.
2.2
L122 Instead of significantly use significant.
Response: Thanks. It was corrected.
3.1
L141 Consider using hours and minutes instead of hour decimals.
Response: I don’t think so. Because it convenient to wright in text and to draw a plot (e.g., Fig.3b) .
L143 I strongly recommend you compare some of this rainfall durations with mean observed historical values, representative of the region. Also to provide some estimates of the future rainfall conditions, based on findings on climate change for the region.
Response: It is better to compare the history data to the current data. However, no history data was found. However, a simple discussion was given based on the published papers.
3.2
L151-152 The sentence “Mean annual Hs were less than 503.11 mm on most of the plots, which were significantly less than those on the bare and cultivated plots 1, 3, 4, 6, 21, and 22” makes no sense, consider rewriting with a better explanation.
Response: I checked the data, and found it was probably wrong. Referring to Fig.4a, the figure 503.11 was replaced by 18.03 in the revised manuscript.
L167 “that these kinds of lands” how much of the region in hectares or km2 occupy these lands? I suggest you to provide a short analysis on the land use / land cover of the region, especially in relation to the “lands” of interest.
Response: The areas of these lands were given, and a short analysis was given in the revised manuscript.
L170 high or higher?
Response: Yes. It was replaced by “high”.
3.3
L183 Consider substituting “which are much larger than the one in this study” by “characterize soils steeper that the ones used in this study”.
Response: According to this suggestion. The phrase was edited.
L208 Consider substituting “made soil crust easily forms” by “resulted in forming soil crusts on the bare plots' surface”.
Response: If it is changed, the meaning of this sentence changed. Therefore, it was not changed.
3.5
L261 based on or upon
Response: Thanks. The word “on” was added.
L263 this needs to be verified also with the amount of rainfall in catchment scale. If the rainfall amount the basin receives has decreased from 2000, then rainfall - and not the adopted measures - is the reason of reduced downstream discharge. Check the historical rainfall amounts to be sure of this suggestion. If verified, then use the finding. Now, it looks rather as an arbitrary conclusion.
Response: Good suggestion. Former studies such as the cited papers after this sentence have found that the decreased discharge can be explained by the decreased precipitation and human activities. However, those studies are conducted based a catchment scale. The present study further verified the given result through analyzing the effect of soil conservation measures on a slope scale. This sentence was edited as “. ….which agrees with the former study results at a catchment scale.
L281 Correct to “In consistence”
Response: Thanks. In consistence was used in the revised manuscript.
L294 Consider using another verb instead of “destroyed”
Response: The word interfere was used to replace this word in the revised manuscript.
5 Conclusions
The fact that the study presents a number of valuable measurements and data analysis, makes documenting an elaborate "conclusions" section, very important. This section needs to be rewritten to support much of the hypothesis.
L317 I recommend you provide here the findings of your study that support this hypothesis.
Response: Yes. The findings that SLRs were higher than 2541 t km-2 yr-1 were added.
L319 I recommend you suggest measures to replace the contour tillage and provide evidence of your study to support this hypothesis here.
Response: Terrace or vegetation strips were suggested to replace contour tillage measure on steep slopes, and an evidence was given using the presented Fig.4b in the manuscript.
Table 1
Specify whether all the treatments were applied for all six years.
Use Italics in latin plant names (e.g. Vitex negundo).
Specify in the manuscript the method you used to calculate vegetation cover percentage in the plots.
Response: Yes. Notes were given. All the treatments were applied for all seven years, the latin plant names were used in Italics, and the method to obtain vegetation coverage in the plots were given in the notes.
Table 2
Correct to amount, not amouint
Consider changing the sentence “Percent of mean annual erosive rainfall amouint to the mean annual one” to “Percent of mean annual erosive rainfall amount in relation to the mean annual”
Response: Thanks. The word “amouint” was corrected.
This error for the phrase was also pointed out, but he or she pointed that it should be “…..annual mean” which is different from this comment on “…mean annual”. In my opinion, both are allowed. However, I think, P (rainfall amount) should be added. Therefore, the phrase was corrected as “Percent of mean annual erosive rainfall amount to the mean annual P”.
Table 3
I recommend you provide all the available information in Table 3, such as 1. the cumulative annual, 2. the mean annual, 3. the event maximum and minimum both for water depth and SLR. This will collect the data in one place and assist the reader to better cope with the provided analysis in the manuscript.
Response: Yes, as suggested, more information was added into Table 3
Figure 3 caption
Provide the information that rainfall incidents data were recorded between 2014 - 2020.
Figure 6b caption
It is m, not mm
Response: Yes. It was corrected.
Round 2
Reviewer 1 Report
The authors have addressed almost all my comments. There is only one comment missing the one related to the sp (specie). The name of all species must be in italics, it is mandatory according to the format.
Reviewer 2 Report
Accepted in the present form